# Effective Thoracoabdominal Pain Management Using Dual Epidural Catheter Placement in Esophageal Reconstruction: A Case Report

**DOI:** 10.3390/reports8040223

**Published:** 2025-10-31

**Authors:** Elizabete Svareniece-Karjaka, Anna Junga, Aleksandrs Malašonoks, Agnese Ozoliņa

**Affiliations:** 1Department of Anesthesiology, Riga East Clinical University Hospital, Hipokrata Street 2, LV-1038 Riga, Latvia; anna.junga@aslimnica.lv (A.J.); agnese.ozolina@aslimnica.lv (A.O.); 2Department of Morphology, Riga Stradiņš University, LV-1007 Riga, Latvia; 3Department of Surgical Oncology, Riga East Clinical University Hospital, LV-1038 Riga, Latvia; aleksandrs.malasonoks@aslimnica.lv; 4Department of Anesthesiology, Intensive Care and Clinical Simulation, Riga Stradiņš University, LV-1007 Riga, Latvia

**Keywords:** dual epidural catheter analgesia, esophageal reconstruction, thoracoabdominal surgery, opioid-sparing multimodal analgesia, case report

## Abstract

**Background and Clinical Significance**: Effective postoperative pain management is crucial in patients undergoing extensive thoracoabdominal surgery, such as esophageal reconstruction, where both thoracic and abdominal incisions are involved. In such cases, a single epidural catheter may not provide sufficient analgesic coverage. Dual epidural analgesia (DEA) offers a potential solution, allowing segmental, targeted pain control while minimizing systemic opioid exposure. **Case Presentation**: A 64-year-old male underwent esophageal reconstruction using a combined thoracoabdominal approach. Two epidural catheters were placed at Th5/6 and Th11/12 levels. Intraoperatively, segmental bupivacaine boluses and multimodal non-opioid intravenous analgesia were administered. Postoperatively, continuous epidural bupivacaine infusion was maintained, supplemented with morphine boluses when the numeric rating scale (NRS) was ≥5. Mean NRS scores were 2 at rest and 5 on movement on postoperative day 1 (POD1); 1 and 4 on POD2; and 3 and 5 on POD3. Total epidural morphine consumption was 36 mg over 340 h, and the 24-h bupivacaine dose was 180 mg (2.77 mg/kg/24 h). No complications were observed. **Conclusions**: Dual epidural analgesia provided effective, opioid-sparing multimodal pain control in complex thoracoabdominal surgery. This case highlights DEA as a safe and feasible approach when single-catheter coverage is inadequate, supporting enhanced recovery and reduced opioid use after esophageal reconstruction.

## 1. Introduction and Clinical Significance

Epidural analgesia (EA) is a standard technique for pain control in major thoracic and abdominal surgery, reducing opioid use and improving recovery. Typically, a single catheter is placed to target the primary surgical area [1].

In complex procedures involving multiple regions, such as combined thoracoabdominal operations, dual epidural analgesia (DEA) may provide broader segmental coverage. Though rarely reported, DEA can achieve effective analgesia across non-contiguous dermatomes when a single catheter is insufficient. Li et al. described improved pain control and fewer opioid-related complications in abdominal surgery [2], while Edelson et al. successfully applied dual thoracic and caudal catheters for abdominoperineal resection [3].

Despite promising results, DEA remains limited to isolated case reports. Although technically more demanding and carrying potential risks of local anesthetic toxicity, DEA may be justified in selected patients. This report describes such a case, highlighting its feasibility and benefits.

## 2. Case Presentation

### 2.1. Primary Surgery: Initial Procedure for Gastroesophageal Junction Carcinoma

A 64-year-old male (65 kg, BMI 22.5 kg/m^2^) underwent a left-sided thoracoabdominal incision in January 2025 for gastroesophageal junction carcinoma. Surgery included partial esophageal and gastric cardia resection with gastroplasty. Due to dysphagia and malnutrition risk, a jejunostomy had been placed five months earlier. He had also received four cycles of palliative chemotherapy before this operation. Preoperatively, he was classified as ASA Physical Status II. Surgery was performed under general anesthesia with thoracic epidural analgesia. Postoperatively, complete esophageal anastomotic dehiscence developed. A drainage tube was inserted into the esophageal stump for continuous aspiration, and a vacuum-assisted closure (VAC) system applied over the thoracotomy wound. The course was further complicated by sepsis and recurrent left-sided empyema. Nevertheless, the patient gradually improved. After functional recovery and rehabilitation, two-level reconstructive surgery with colonic interposition for esophageal reconstruction was scheduled three months later.

### 2.2. Secondary Surgery: Esophageal Reconstruction with Colonic Interposition Under General Anesthesia Using Dual Epidural Analgesia

Preoperative assessment classified the patient as ASA Physical Status IV, reflecting a severe systemic condition related to recent sepsis, recurrent pneumonia, and the underlying oncologic disease. The patient was also at increased risk of postoperative acute and chronic pain due to prolonged opioid exposure, having received daily intravenous pethidine hydrochloride 50 mg (Doloblok^®^ 50 mg/mL, Sanitas, Kaunas, Lithuania) during the preceding month. This likely contributed to opioid tolerance and opioid-induced hyperalgesia, thereby reducing the effectiveness of postoperative analgesia. These factors were key considerations in selecting an opioid-sparing anesthetic strategy.

There were substantial concerns that a single epidural catheter might not provide adequate analgesic coverage, as the procedure involved both a midline laparotomy and a right thoracotomy at the fifth intercostal space. Effective coverage for such thoracoabdominal surgery typically requires blockade from T4 to L2 (8–10 dermatomes), whereas a single catheter generally covers only 4–6. Increasing the dose at one level could cause excessive spread and hemodynamic instability. Therefore, a dual epidural approach (T5/6 and T11/12) was selected to achieve targeted analgesia with lower anesthetic doses and minimal systemic effects.

DEA catheters (B. Braun Perifix^®^, Melsungen, Germany) were inserted preoperatively in the sitting position using an interlaminar approach. Levels were identified by palpation and thoracic spine X-ray. The epidural space was found with a saline-air technique at Th5/6 (4.5 cm depth) for esophageal coverage and at Th11/12 (5 cm depth) for colonic coverage (Figure 1). The catheters were advanced 6 cm, covering Th3–8 and Th9–L2 dermatomes. Initially, 5 mL boluses of 0.25% bupivacaine hydrochloride (Bupivacaine-Grindeks Spinal^®^, Grindex, Riga, Latvia) were administered at Th11/12, and 20 min before the thoracic incision, through the Th5/6 catheter. Subsequently, 5 mL boluses of 0.25% local anesthetic were administered via both catheters every 90 min.

Surgery was performed under general anesthesia with sevoflurane (Sevorane^®^, AbbVie Inc., Dublin, Ireland) at MAC 0.7–0.9. Neuromuscular relaxation was maintained with cisatracurium (Cisatracurium-Kalceks^®^, 2 mg/mL, Kalcex, Riga, Latvia). Multimodal nonopioid intravenous analgesia included paracetamol 1000 mg (Supofen^®^, Laboratórios Basi, Mortágua, Portugal) and metamizole 1000 mg (Metamizole sodium-Kalceks^®^, Kalcex, Riga, Latvia) every 6 h, plus single doses of ketorolac 30 mg (Ketanov^®^, Sun Pharmaceutical Industries, Hoofddorp, The Netherlands) and dexamethasone 8 mg (Dexamethasone-Kalceks^®^, Kalcex, Riga, Latvia) after induction. The 530-min surgery required fentanyl 0.2 mg or 1.54 mcg/kg (Fentanyl citrate-Kalceks^®^, Kalcex, Riga, Latvia) at induction and before abdominal incision. Norepinephrine 0.0205–0.0820 mcg/kg/min (Norepinephrine Kabi^®^, Fresenius Kabi, Kutno, Poland) was infused to maintain MAP > 65 mmHg, as mild hemodynamic fluctuations were observed during the surgery.

Postoperatively, the patient was transferred to the Intensive Care Unit (ICU), where a continuous epidural infusion of 0.125% bupivacaine at 3 mL/h was maintained through both catheters. Two hours after surgery, following tracheal extubation, the Bromage score was 0 and NRS pain score was 0/10 at rest.

### 2.3. Postoperative Pain Management

During the first postoperative days, pain at rest ranged from 1 to 3 on the NRS, and pain on movement or coughing from 3 to 6. Epidural boluses of morphine 3 mg (Morphine hydrochloride-Kalceks^®^, 10 mg/mL, Kalcex, Riga, Latvia) were given on average once daily when NRS ≥ 5, as shown in Figure 2. Total epidural morphine consumption was 36 mg over 340 h, avoiding intravenous opioids. The 24-h bupivacaine dose was 180 mg (2.77 mg/kg/24 h). Additional analgesia included paracetamol 1000 mg three times daily, metamizole 1000 mg three times daily, and dexketoprofen 50 mg twice daily (Dolmen^®^ 50 mg/2 mL, Berlin-Chemie Menarini, Florence, Italy).

Enhanced monitoring was provided due to the cumulative risk of local anesthetic systemic toxicity (LAST). The protocol included frequent assessment of hemodynamics, respiratory function, neurological status (motor and sensory block), pain scores, and early signs of LAST such as perioral numbness or changes in mentation.

No significant hemodynamic instability occurred despite dual catheters and cumulative anesthetic dose. The patient remained neurologically intact and mobilized early: he could sit, stand, and perform in-bed physical activity with supervision, with both catheters in place. Prophylactic enoxaparin 0.4 mL (Inhixa^®^, Techdow Pharma Spain S.L., Madrid, Spain) was given subcutaneously once daily.

DEA catheters were removed on POD14 after transfer from the ICU to the surgical ward. A relaparotomy was performed on POD7 for suspected colonic necrosis on computed tomography, but the colon was viable at revision. The catheters were retained to maintain analgesia during ongoing ICU care.

At discharge, no DEA-related complications were observed.

## 3. Discussion

The use of DEA for combined thoracoabdominal procedures is rarely reported. Although technically demanding, DEA enables precise segmental analgesia when single-catheter coverage is insufficient [4]. In this case, DEA at Th5/6 and Th11/12 provided targeted intra- and postoperative pain control during colonic interposition for esophageal reconstruction.

A large retrospective study by Sarridou et al. demonstrated that thoracic epidural analgesia is associated with minimal major complications in more than 1100 patients [5]. Similarly, our patient achieved effective analgesia without complications. However, evidence on DEA remains limited to case reports and small series. The main concern is local anesthetic systemic toxicity (LAST) due to drug accumulation from two active catheters, underscoring the importance of dose monitoring [6].

The patient received 180 mg of bupivacaine (2.77 mg/kg/24 h), well within the recommended limit of 400 mg/24 h [7]. Pain control at rest was satisfactory, though movement-related NRS values ≥ 5 suggested that slightly higher doses might have improved dynamic analgesia.

Epidural morphine (3 mg) was administered within the adult dosing range (2–5 mg). Despite its potential to delay bowel recovery, continuous epidural analgesia—even when combined with morphine—promotes faster gastrointestinal function compared to systemic opioids [8,9]. During the 14-day ICU stay, the patient received daily doses of 3000 mg paracetamol, 3000 mg metamizole, and 100 mg dexketoprofen. No morphine-related or local anesthetic adverse effects were observed.

Alternative approaches, such as combining lumbar epidural with thoracic fascial plane blocks (e.g., erector spinae plane or paravertebral), have been reported, but these techniques carry a higher risk of catheter migration and limited evidence in esophageal reconstruction [10]. Both epidural catheters remained functional and were safely removed on POD 14 after relaparotomy.

These findings support DEA as a valuable adjunct for complex thoracoabdominal procedures, particularly when ultrasound guidance is used for epidural placement. A recent randomized trial demonstrated higher first-pass success and fewer punctures with ultrasound compared to landmark-based techniques [11]. Both punctures in this case were successful on the first attempt using anatomical landmarks and spinal X-ray, although ultrasound may further enhance precision and safety.

Aside from the risk of LAST, DEA may increase the likelihood of infection, catheter migration, or hemodynamic instability from extended sympathetic blockade. The technique is technically demanding, requiring experienced staff, careful planning, and patient selection. Strict aseptic technique, daily inspection, and cautious dose titration are essential to minimize risks and ensure safety.

Prolonged use of two epidural catheters inherently increases the risk of infection, dislodgement, or occlusion, requiring vigilant monitoring and meticulous adherence to aseptic protocols.

Guidelines from the American Society of Regional Anesthesia and Pain Medicine (ASRA), the European Society of Anaesthesiology and Intensive Care (ESAIC), and the United Kingdom National Institute for Health and Care Excellence (NICE) recommend removing epidural catheters within 3–5 days once no longer clinically required to reduce infection risk. However, extended use may be justified when strict aseptic care, daily assessment, and documentation are maintained. In this case, all precautions were followed, including daily inspection and dressing changes, and prolonged catheterization was necessary due to a relaparotomy on POD 7.

Similarly to this case, Brown et al. [12] reported that the dual-catheter technique in Ivor-Lewis esophagectomy improved dynamic analgesia and reduced opioid use compared with a single epidural, without added complications. These findings further support DEA as a feasible and effective approach in extensive thoracoabdominal surgery requiring broader dermatomal coverage.

## 4. Conclusions

Dual epidural analgesia provided stable and prolonged pain control, reduced opioid requirements, and supported early mobilization. Consistent with published evidence, it enabled effective thoracoabdominal coverage without hemodynamic or catheter-related complications. Although guidelines advise limiting epidural catheterization to 3–5 days, careful daily assessment allowed safe prolonged use in this case. DEA appears to be a feasible, safe, and effective option for selected patients in complex thoracoabdominal surgery when single-catheter coverage is inadequate.

## Figures and Tables

**Figure 1 reports-08-00223-f001:**
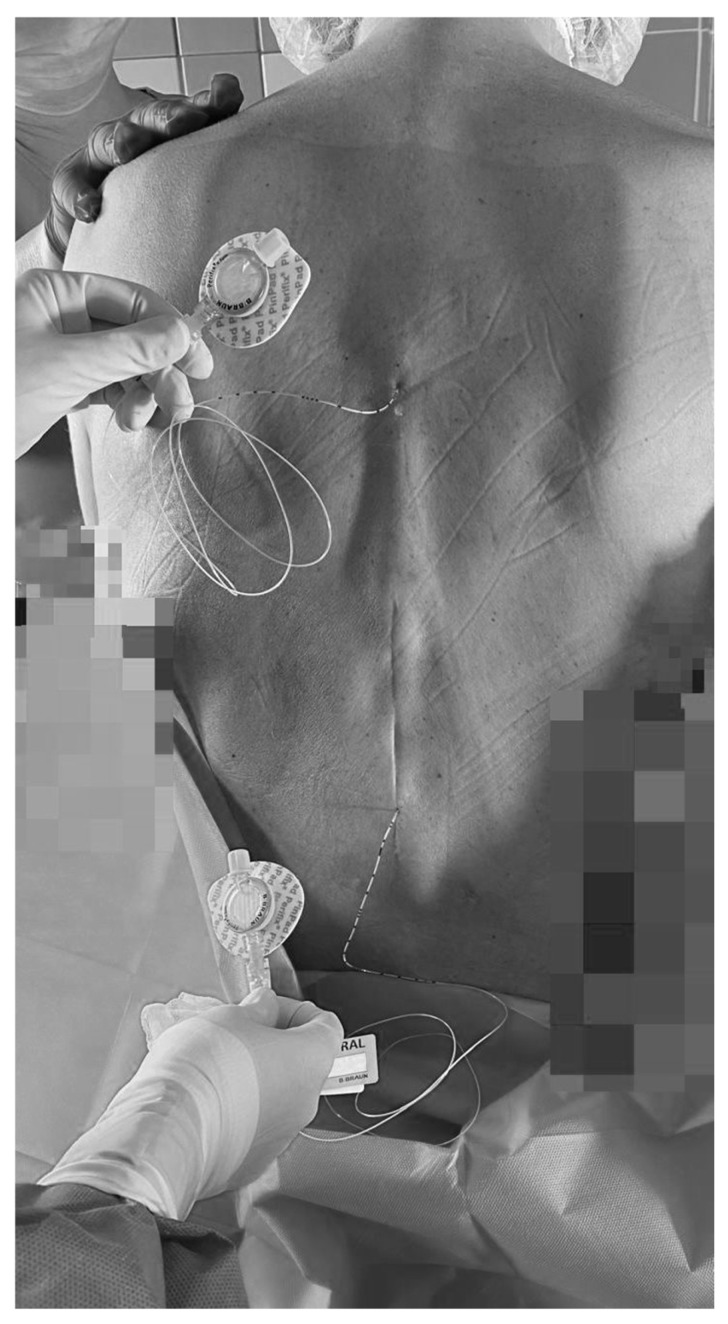
Dual epidural catheters placed at Th5/6 and Th11/12 using the saline–air technique (depths 4.5 cm and 5 cm) to provide segmental coverage for esophageal and colonic surgery. Both catheters were secured with sterile dressings and labeled for continuous perioperative infusion.

**Figure 2 reports-08-00223-f002:**
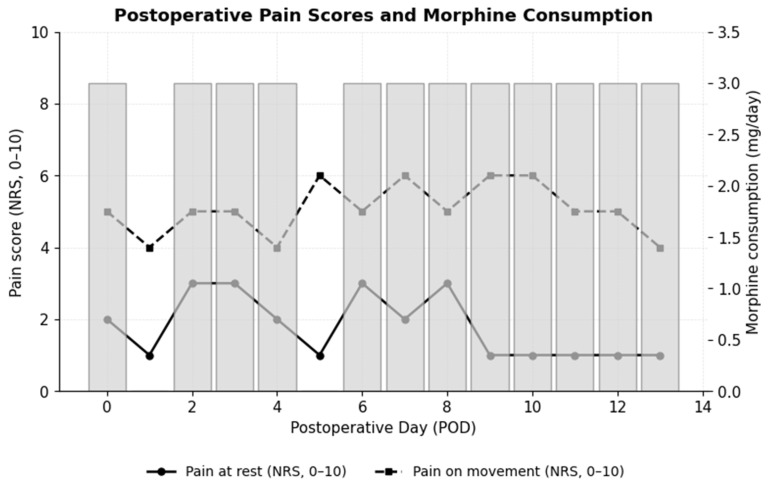
Postoperative pain scores at rest and on movement (NRS, 0–10) and daily epidural morphine consumption (mg/day) during the first 14 postoperative days. Pain scores are presented as mean values. Solid line with circles represents pain at rest; dashed line with squares represents pain on movement; vertical bars represent daily morphine consumption. Lines show NRS pain scores at rest and on movement, while bars indicate daily morphine use. POD—postoperative day; NRS—numeric rating scale.

## Data Availability

The original data presented in the study are included in the article, further inquiries can be directed to the corresponding author.

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
