# Peer review of "Effective Thoracoabdominal Pain Management Using Dual Epidural Catheter Placement in Esophageal Reconstruction: A Case Report"

_reports, 2025, doi:10.3390/reports8040223_

Round 1
Reviewer 1 Report
Comments and Suggestions for Authors
The case report is very interesting, even if some data needs more clarity.
ASA classification is not related to postoperative complications, bu t maybe to the septic state as well as the oncological state.
It is not so clear the link of risk of postsurgical acute and chronic pain from prolonged opioid exposure, please better explain.
Why bupivacaine was chosen and not other LA with less cardiovascular impact? And whty Norepinep rine 0.0205–0.0820 mcg/kg/min was infused?
How many hours did surgery last?Were other pain killer given? How many times? And why the epidural infusion was continued for 14 days, at same doses?
Moreover, please discuss the choice of low concentration in both catheter of AL in postop periodo as well as small boluses in intraop. Could the same pian control be obtained with 1 single epidural cathether positioned at middle-thoracic level with higher boluses and higher concentration? Are there any other reports in literature about esophageal reconstruction managed with epidural or spinal approach?
Author Response
Comments 1: The case report is very interesting, even if some data needs more clarity.
Response 1: Thank you very much for your positive feedback and for finding our case report of interest. We sincerely appreciate your constructive comments and will revise the manuscript to improve clarity where needed.
Comments 2: ASA classification is not related to postoperative complications, bu t maybe to the septic state as well as the oncological state.
Response 2: We thank the reviewer for this valuable comment and fully agree with the observation. The corresponding correction has been made in the revised manuscript and is highlighted in red in lines 58-60.
Comments 3: It is not so clear the link of risk of postsurgical acute and chronic pain from prolonged opioid exposure, please better explain.
Response 3: Prolonged preoperative opioid exposure may contribute to the development of both acute and chronic postsurgical pain due to opioid tolerance and opioid-induced hyperalgesia, which can alter pain processing and reduce the effectiveness of postoperative analgesia. The correction can be seen in the revised manuscript, highlighted in red in lines 63 and 64.
Comments 4: Why bupivacaine was chosen and not other LA with less cardiovascular impact? And whty Norepinep rine 0.0205–0.0820 mcg/kg/min was infused?
Response 4: We thank the reviewer for this relevant question. Bupivacaine was selected because of its high potency, long duration of action, and well-established efficacy for epidural analgesia, particularly in thoracic and abdominal procedures requiring dense sensory block. Although other local anesthetics, such as ropivacaine or levobupivacaine, may have a lower cardiovascular impact, bupivacaine remains the most readily available and routinely used agent in our institution, with extensive clinical experience supporting its safety when used at low concentrations and with appropriate monitoring.
Regarding the use of norepinephrine (0.0205–0.0820 µg/kg/min), it was administered in low doses to maintain a mean arterial pressure (MAP) above 65 mmHg during epidural-induced sympathetic blockade. This hemodynamic management was consistent with standard perioperative practice to ensure adequate organ perfusion and cardiovascular stability. The addition can be found in the revised manuscript, highlighted in red in lines 93 and 94.
Comments 5: How many hours did surgery last?Were other pain killer given? How many times? And why the epidural infusion was continued for 14 days, at same doses?
Response 5: The surgical procedure lasted 8 hours and 50 minutes (530 minutes), as mentioned in line 90 of the case report.Intraoperatively, following induction, multimodal non-opioid intravenous analgesia was administered, including paracetamol 1000 mg, metamizole 1000 mg, and ketorolac 30 mg. Six hours later, the patient received additional paracetamol 1000 mg and metamizole 1000 mg intravenously. The multimodal intraoperative management is described in lines 85–90 of the case report.
Postoperatively, paracetamol 1000 mg three times daily, metamizole 1000 mg three times daily, and dexketoprofen 50 mg twice daily were administered, in addition to continuous epidural bupivacaine infusion, as described in lines 109 and 110 of the text.
The infusion was maintained for 14 days because the patient underwent a relaparotomy on postoperative day 7. The bupivacaine dose was not modified, as the patient remained comfortable, reported adequate analgesia, and was able to participate actively in the rehabilitation process. However, as indicated in the Discussion section, highlighted in green from lines 146 to 148, an addition has been made stating that, retrospectively, based on the NRS scores during coughing, we believe the bupivacaine dose should have been increased.
Comments 6: Moreover, please discuss the choice of low concentration in both catheter of AL in postop periodo as well as small boluses in intraop. Could the same pian control be obtained with 1 single epidural cathether positioned at middle-thoracic level with higher boluses and higher concentration? Are there any other reports in literature about esophageal reconstruction managed with epidural or spinal approach?
Response 6: This was our first case using two epidural catheters simultaneously. Low concentrations of local anesthetic and small intraoperative boluses were chosen to minimize the risk of systemic toxicity. Low-dose norepinephrine was required to maintain hemodynamic stability, indicating adequate sympathetic block. Postoperatively, the same low-dose infusion was continued, as the patient remained comfortable and able to participate in rehabilitation. In future similar cases, we would consider increasing the local anesthetic dose to further optimize analgesia while maintaining stability, we have added this clarification to the text, which can be found in the Discussion section, in lines 146–148, where we state that, retrospectively, the local anesthetic dose should have been increased.
A single epidural catheter would likely not have provided adequate analgesia, as the procedure involved both a midline laparotomy and a right thoracotomy at the fifth intercostal space. Effective coverage for such thoracoabdominal surgery typically requires blockade from T4 to L2 (8–10 dermatomes), while a single catheter usually covers only 4–6 dermatomes. Increasing the dose at a single level could result in excessive spread and hemodynamic instability. Therefore, a dual epidural approach (T5/6 and T11/12) was chosen to ensure targeted, segmental analgesia with lower anesthetic doses and minimal systemic effects. The addition can be seen in the text from line 66 to line 73, highlighted in red.
Yes, relevant literature about esophageal reconstruction managed with epidural or spinal approaches exists.
Epidural analgesia has been shown to be beneficial not only during esophagectomy but also in the reconstruction phase. A recent review emphasized its role in reducing the surgical stress response and improving clinical outcomes. Similarly, Kaufmann et al. reported the routine use of epidural analgesia in patients undergoing esophagectomy, including 4% who required colon interposition for reconstruction, supporting its applicability in such cases. Reports describing the use of spinal or combined spinal–epidural techniques are limited and primarily focus on intrathecal morphine as an adjunct for postoperative analgesia, showing effective pain control when appropriate monitoring for delayed respiratory depression is ensured. Additionally, we have expanded the Discussion section (lines 179–183, highlighted in green) to include a comparison of our findings with a larger study evaluating single versus dual epidural catheter placement in patients undergoing Ivor-Lewis esophagectomy.
Reports of dual epidural catheters exist but remain limited, while evidence for spinal or combined spinal–epidural techniques is scarce and confined to isolated case reports.
I would like to inform you that we have substantially revised and condensed the Discussion section in accordance with the reviewers’ recommendations and suggestions. In addition, we have also made extensive modifications to the Conclusion section to ensure consistency with the revised Discussion.
Please find attached the Word file in which all revisions are highlighted.

Reviewer 2 Report
Comments and Suggestions for Authors
This is a well-structured and informative case report describing the successful use of dual epidural analgesia (DEA) in complex thoracoabdominal esophageal reconstruction. The manuscript clearly outlines the rationale for DEA, provides comprehensive perioperative details, and supports its conclusions with appropriate literature. The case contributes valuable clinical evidence regarding the feasibility, safety, and opioid-sparing benefits of this rarely used technique.
Overall, the manuscript is of good quality, but several minor improvements are recommended before publication:
1. Clarify the rationale for selecting dual epidural catheters instead of a single epidural or alternative regional blocks (e.g., paravertebral or erector spinae plane block).
2. Streamline the Discussion section by removing repetitive statements, particularly those related to postoperative outcomes and analgesic effects.
3. Improve the resolution and labeling of Figure 2 to ensure clarity and consistency with journal standards.
4. A brief commentary on the limitations and risk management of DEA (infection, catheter migration, local anesthetic toxicity) would further strengthen the clinical relevance.
Once these revisions are addressed, the manuscript will be suitable for publication.
Comments on the Quality of English LanguageThe English is clear, grammatically correct, and professionally written. The manuscript reads smoothly and conveys the clinical content effectively.
Only minor editorial polishing may be required to improve conciseness and flow, but no major language revision is needed.
Author Response
Thank you very much for your positive feedback and for finding our case report of interest. We sincerely appreciate your constructive comments and will revise the manuscript to improve clarity where needed.
Comments 1: Clarify the rationale for selecting dual epidural catheters instead of a single epidural or alternative regional blocks (e.g., paravertebral or erector spinae plane block).
Response 1: A single epidural catheter would likely not have provided adequate analgesia, as the procedure involved both a midline laparotomy and a right thoracotomy at the fifth intercostal space. Effective coverage for such thoracoabdominal surgery typically requires blockade from T4 to L2 (8–10 dermatomes), while a single catheter usually covers only 4–6 dermatomes. Increasing the dose at a single level could result in excessive spread and hemodynamic instability. Therefore, a dual epidural approach (T5/6 and T11/12) was chosen to ensure targeted, segmental analgesia with lower anesthetic doses and minimal systemic effects. The addition can be found in the text, highlighted in red, in lines 66–73.
Epidural catheters were also selected because they allowed for continuous and reliable postoperative analgesia. Considering that peripheral catheter techniques (such as paravertebral or erector spinae plane blocks) have a higher risk of catheter dislocation, as reported in the literature and discussed in our manuscript, the dual epidural approach was considered the most stable and effective option for this complex thoracoabdominal reconstruction. We tried to address this in the Discussion section, in lines 155–158.
Comments 2: Streamline the Discussion section by removing repetitive statements, particularly those related to postoperative outcomes and analgesic effects.
Response 2: Thank you for the suggestion. We have shortened and streamlined the Discussion section to make it more concise and focused, as reflected in the revised text from lines 133 to 183.
Comments 3: Improve the resolution and labeling of Figure 2 to ensure clarity and consistency with journal standards.
Response 3: Thank you for the valuable suggestion. The revisions have been made to improve the figure’s resolution and labeling for better clarity and consistency with journal standards. The changes can be found in the text, highlighted in yellow, in lines 112–118.
Comments 4: A brief commentary on the limitations and risk management of DEA (infection, catheter migration, local anesthetic toxicity) would further strengthen the clinical relevance.
Response 4: Thank you for this valuable comment. Information regarding the risk of local anesthetic systemic toxicity (LAST) and its monitoring has been included in the Discussion section, in lines 142–144. Additional limitations of the technique have been added to the end of the Discussion section, in lines 166–170, highlighted in yellow.
I would like to inform you that we have substantially revised and condensed the Discussion section in accordance with the reviewers’ recommendations and suggestions. In addition, we have also made extensive modifications to the Conclusion section to ensure consistency with the revised Discussion.
Please find attached the Word file in which all revisions are highlighted.

Reviewer 3 Report
Comments and Suggestions for Authors
Overall Assessment
This case report addresses a clinically relevant and rarely described technique—dual epidural catheter analgesia for complex thoracoabdominal surgery. The manuscript is clearly written and informative; however, several important issues should be clarified to strengthen its scientific value. In particular, the prolonged catheter duration, suboptimal pain control during movement, incomplete reporting of intravenous analgesic doses, and lack of comparative data with single-catheter epidural analgesia warrant further discussion.
Specific Comments
- The duration of dual epidural catheterization (14 days) is considerably longer than current guideline recommendations. Most professional societies (e.g., ASRA, NICE) suggest that epidural catheters be maintained for no more than 3–5 days to reduce infection risk. The authors should briefly discuss this issue, clarify the rationale for prolonged catheterization, and reference relevant guideline recommendations.
- The reported NRS values during movement were frequently ≥5 throughout the postoperative period, implying that analgesia may have been inadequate during mobilization or coughing. The authors should discuss this issue.
- While the manuscript lists the types of intravenous non-opioid analgesics administered, it does not provide the cumulative doses or duration of administration. Detailed reporting of total daily and overall doses of paracetamol, metamizole, dexketoprofen, and any other adjuncts would enable readers to more accurately assess the adequacy of multimodal analgesia and the true opioid-sparing impact of the dual epidural technique.
- To enhance external validation, the authors should compare their findings with published data on patients undergoing similar thoracoabdominal procedures managed with single epidural catheter analgesia. Including opioid consumption, pain scores, and complication rates from previous studies would help readers understand the incremental benefit of the dual-catheter approach.
Author Response
Thank you very much for your positive feedback and for finding our case report of interest. We sincerely appreciate your constructive comments and will revise the manuscript to improve clarity where needed.
Comments 1: The duration of dual epidural catheterization (14 days) is considerably longer than current guideline recommendations. Most professional societies (e.g., ASRA, NICE) suggest that epidural catheters be maintained for no more than 3–5 days to reduce infection risk. The authors should briefly discuss this issue, clarify the rationale for prolonged catheterization, and reference relevant guideline recommendations.
Response 1: Thank you for the comment. We have revised the Discussion section in accordance with this recommendation. The changes can be seen in the text, highlighted in green, in lines 171–178.
Comments 2: The reported NRS values during movement were frequently ≥5 throughout the postoperative period, implying that analgesia may have been inadequate during mobilization or coughing. The authors should discuss this issue.
Response 2: We thank the reviewer for this valuable observation. The corresponding clarification has been added and can be found in the revised manuscript, highlighted in green in lines 146–148.
Comments 3: While the manuscript lists the types of intravenous non-opioid analgesics administered, it does not provide the cumulative doses or duration of administration. Detailed reporting of total daily and overall doses of paracetamol, metamizole, dexketoprofen, and any other adjuncts would enable readers to more accurately assess the adequacy of multimodal analgesia and the true opioid-sparing impact of the dual epidural technique.
Response 3: Thank you for the suggestion. The addition can be seen in the text, highlighted in green, in lines 152–153.
Comments 4: To enhance external validation, the authors should compare their findings with published data on patients undergoing similar thoracoabdominal procedures managed with single epidural catheter analgesia. Including opioid consumption, pain scores, and complication rates from previous studies would help readers understand the incremental benefit of the dual-catheter approach.
Response 4: Thank you for the valuable suggestion. We have expanded the Discussion section to include a comparison with published data on single epidural catheter analgesia. The corresponding additions can be found in lines 179–183, highlighted in green.
I would like to inform you that we have substantially revised and condensed the Discussion section in accordance with the reviewers’ recommendations and suggestions. In addition, we have also made extensive modifications to the Conclusion section to ensure consistency with the revised Discussion.
Please find attached the Word file in which all revisions are highlighted.

Round 2
Reviewer 1 Report
Comments and Suggestions for Authors
The paper is improved. I suggest to better underline the risk of dual catheter insertion and management until 14 days.
Author Response
Comments 1: The paper is improved. I suggest to better underline the risk of dual catheter insertion and management until 14 days.
Response1: Thank you very much for your valuable comment. We fully agree that the risks associated with dual epidural catheter insertion and prolonged management (up to 14 days) should be emphasized. Accordingly, we have added a clarification in the Discussion section (lines 171–173, highlighted in pink).

Reviewer 3 Report
Comments and Suggestions for Authors
Thank you for the response. I have no further comments.
Author Response
Comments1: Thank you for the response. I have no further comments.
Response1: Thank you very much for your kind feedback and for taking the time to review our manuscript. We sincerely appreciate your valuable input throughout the process.